# Antibody Profiling of Microbial Antigens in the Blood of COVID-19 mRNA Vaccine Recipients Using Microbial Protein Microarrays

**DOI:** 10.3390/vaccines11111694

**Published:** 2023-11-07

**Authors:** Hiroaki Saito, Hiroki Yoshimura, Makoto Yoshida, Yuta Tani, Moe Kawashima, Taiga Uchiyama, Tianchen Zhao, Chika Yamamoto, Yurie Kobashi, Toyoaki Sawano, Seiya Imoto, Hyeongki Park, Naotoshi Nakamura, Shingo Iwami, Yudai Kaneko, Aya Nakayama, Tatsuhiko Kodama, Masatoshi Wakui, Takeshi Kawamura, Masaharu Tsubokura

**Affiliations:** 1Department of Radiation Health Management, Fukushima Medical University School of Medicine, Fukushima, Fukushima 960-1247, Japan; 2Department of Internal Medicine, Soma Central Hospital, Soma, Fukushima 976-0016, Japan; 3School of Medicine, Hiroshima University, Hiroshima, Hiroshima 739-8511, Japan; 4Faculty of Medicine, Teikyo University School of Medicine, Itabashi-ku, Tokyo 173-8605, Japan; 5Medical Governance Research Institute, Minato-ku, Tokyo 108-0074, Japan; 6Department of Laboratory Medicine, Keio University School of Medicine, Shinjuku-ku, Tokyo 160-8582, Japan; 7Department of Internal Medicine, Serireikai Group Hirata Central Hospital, Ishikawa County, Fukushima 963-8202, Japan; 8Division of Health Medical Intelligence, Human Genome Center, Institute of Medical Science, The University of Tokyo, Shirokanedai, Minato-ku, Tokyo 108-8639, Japan; 9Interdisciplinary Biology Laboratory (iBLab), Division of Natural Science, Graduate School of Science, Nagoya University, Chikusa-ku, Nagoya 464-8601, Japaniwamishingo@gmail.com (S.I.); 10Medical & Biological Laboratories Co., Ltd., Minato-ku, Tokyo 105-0012, Japan; 11Laboratory for Systems Biology and Medicine, Research Centre for Advanced Science and Technology (RCAST), The University of Tokyo, Meguro-ku, Tokyo 153-8904, Japan; 12Isotope Science Centre, The University of Tokyo, Bunkyo-ku, Tokyo 113-0032, Japan; 13Minamisoma Municipal General Hospital, Minamisoma, Fukushima 975-0033, Japan

**Keywords:** vaccine, COVID-19, antibody, bacteria

## Abstract

Although studies have demonstrated that infections with various viruses, bacteria, and parasites can modulate the immune system, no study has investigated changes in antibodies against microbial antigens after the COVID-19 mRNA vaccination. IgG antibodies against microbial antigens in the blood of vaccinees were comprehensively analyzed using microbial protein microarrays that carried approximately 5000 microbe-derived proteins. Changes in antibodies against microbial antigens were scrutinized in healthy participants enrolled in the Fukushima Vaccination Community Survey conducted in Fukushima Prefecture, Japan, after their second and third COVID-19 mRNA vaccinations. Antibody profiling of six groups stratified by antibody titer and the remaining neutralizing antibodies was also performed to study the dynamics of neutralizing antibodies against SARS-CoV-2 and the changes in antibodies against microbial antigens. The results showed that changes in antibodies against microbial antigens other than SARS-CoV-2 antigens were extremely limited after COVID-19 vaccination. In addition, antibodies against a staphylococcal complement inhibitor have been identified as microbial antigens that are associated with increased levels of neutralizing antibodies against SARS-CoV-2. These antibodies may be a predictor of the maintenance of neutralizing antibodies following the administration of a COVID-19 mRNA vaccine.

## 1. Introduction

The coronavirus disease (COVID-19) pandemic, caused by severe acute respiratory syndrome coronavirus 2 (SARS-CoV-2), was declared by the World Health Organization on 11 March 2020 [1]. Since then, vaccine development has been actively pursued worldwide to combat the pandemic. In Japan, the administration of messenger RNA (mRNA) vaccines, namely BNT162b2 (Pfizer Biotech) and mRNA-1273 (Moderna) began in February 2021 and May 2021, respectively [2]. As of May 2023, approximately 86 million people have received three doses of the vaccine in Japan [3]. These mRNA-based vaccines have been instrumental in curtailing the pandemic, demonstrating their efficacy in eliciting both humoral and cellular immune responses against COVID-19, thus mitigating infection rates and the severity of the disease [4,5,6]. However, the degree of induction and maintenance of neutralizing antibodies in response to COVID-19 mRNA vaccines varies among individuals.

In Fukushima Prefecture, Japan, the Fukushima Vaccination Community Survey (FVCS) has been conducted since the early stages of the pandemic to test for antibodies against COVID-19. Through a multi-sectional collaboration between local governments, universities, and hospitals, antibody tests are conducted every three months on approximately 2500 people to investigate adverse reactions after COVID-19 vaccination, and the findings are reported to the local population [7]. Thus, it is possible to evaluate adverse reactions and immunity dynamics after booster vaccination at the regional level based on the information collected during the FVCS. We previously published studies on the effects of vaccination on humoral and cellular immunity as determined through the FVCS [7,8,9,10,11,12]. We analyzed the changes in neutralizing antibody titers and the factors influencing these titers by measuring neutralizing antibodies against SARS-CoV-2 over time after vaccination [7]. It is crucial to evaluate the state of immunity over time post-vaccination using such cohorts.

While COVID-19 vaccines are effective in preventing severe infections, individual differences in antibody formation exist. Moreover, there are variations in long-term antibody retention. Based on the FVCS cohort, mathematical modeling and machine learning have been used to stratify the survey participants into six groups according to the time-course patterns of their neutralizing antibody titer trajectories after their second SARS-CoV-2 vaccination [11]. In this model, G1 represented the “rich” responder group with sustained high antibody titers, while G5 and G6, the “poor” responder groups, exhibited low titers. G2, G3, and G4 were intermediate groups. Several factors have been identified as inhibitors to antibody formation post-vaccination [12]. For example, patients with immunodeficiencies, those on maintenance dialysis, and the elderly, often have reduced antibody responses after SARS-CoV-2 vaccination. However, even among those without such medical conditions, variations in antibody responses exist, and the reasons remain unclear. Identifying populations with diminished vaccine efficacy is vital for determining COVID-19 vaccination priorities and discussing the need for repeated vaccinations.

The relationship between the efficacy of vaccines and infections from other microorganisms has been a topic of discussion [13]. To elucidate the individual differences in the efficacy of the COVID-19 vaccine, we hypothesize that examining antibodies against other microorganisms might offer valuable insight. This perspective gains particular relevance when considering countries like Japan, where a vast majority of the population has been vaccinated against diseases such as measles, rubella, and mumps, and where individual variations in the acquisition of antibodies from these vaccines are documented [14,15,16]. Furthermore, certain pathogens, such as the human immunodeficiency virus, measles virus, Salmonella, Schistosoma, and Nematospiroides, are known to exert immunosuppressive effects [17]. There is speculation on whether the presence of antibodies against these microorganisms, which suggest a current or previous infection, might serve as predictors for the efficacy of other vaccines. Furthermore, there have been instances where vaccines targeting one microorganism exhibit cross-reactivity with antigens from other closely related microorganisms [18]. In addition, the intestinal microbiota plays a pivotal role in immune responses and are thought to be associated with vaccine responses [19]. Prior exposure to microorganisms and subsequent antibody production as a result of the immune response might be associated with the efficacy of vaccines and both the humoral and cellular immune reactions [20]. Despite these speculations and preliminary findings, comprehensive studies delving into the interplay between COVID-19 vaccine efficacy and antibodies against other infectious agents remain scarce.

In this study, we performed IgG antibody profiling against multiple microbial proteins using serum samples from recipients of COVID-19 mRNA vaccines to determine the effects of vaccination on antibodies against microorganisms other than SARS-CoV-2. Our focus was to investigate the relationship between neutralizing antibody production for SARS-CoV-2 and antibody profiles against other microbes.

## 2. Materials and Methods

This study was approved by the ethics committees of Hirata Central Hospital (number 2021-0611-1) and Fukushima Medical University School of Medicine (number 2021-116).

### 2.1. Study Design and Participants

This study included two sets of participants from the FVCS [7]. Based on the post-vaccination questionnaire survey, the patients were confirmed to have no special preexisting medical conditions or medications. All participants were confirmed to be free of SARS-CoV-2 infection by antibody testing against the SARS-CoV-2 nucleoprotein. In the first set, we investigated changes in antibody levels over time, from the second to the third vaccinations. To study the changes in antibodies against microbial antigens in vaccinees over time, healthy individuals with no significant medical history were randomly selected. Eight participants had blood samples taken three times after the second BNT162b2 vaccination (approximately 30, 90, and 180 days after vaccination) and approximately 60 days after the third BNT162b2 vaccination. The samples were analyzed for IgG antibody profiling using microbial protein microarrays.

In the second set, we examined the relationship between the production of neutralizing antibodies against SARS-CoV-2 and the production of antibodies against other microorganisms. To investigate the relationship between the production of neutralizing antibodies against SARS-CoV-2 after COVID-19 vaccination and the production of antibodies against other microorganisms, specimens that were collected approximately 90 days after the second BNT162b2 vaccination were subjected to IgG antibody profiling using microbial microarrays. We selected this timeframe because vaccine-induced antibody levels begin to vary around this period, offering a clearer insight into antibody retention in the population. We analyzed antibody profiles in six groups (G1–6, as previously described) which were stratified based on antibody formation, the temporal persistence of antibodies, and remaining neutralizing antibody levels [11]. Thirty-nine samples were selected for analysis from the six groups stratified by vaccine-elicited time-course antibody dynamics in the FVCS [7,11].

### 2.2. Measurement of IgG Antibody against Nucleoprotein and Neutralizing Activity

We gauged the level of antibodies targeting the SARS-CoV-2 nucleocapsid (N) protein and receptor-binding domain (RBD) using the iFlash 3000 (YHLO Biotech, Shenzhen, China) and its complementary iFlash-2019-nCoV series assay kits (YHLO Biotech). All procedures adhered to the guidelines provided by the manufacturer. A daily quality assurance routine was conducted before any measurements. The threshold levels for anti-N antibodies and RBD were set at 10 arbitrary units per mL (AU/mL).

### 2.3. Microbial Protein Microarray

Fukushima Medical University has developed a protein microarray carrying protein extracts and antigenic proteins from more than 4000 human-related microorganisms [21]. Using this microbial microarray, it is possible to profile antibodies against microorganisms present in the human blood. Microbial protein microarrays were utilized to profile serum IgG, comprising 3437 protein extract samples and 1606 recombinant protein samples from an array of pathogens such as viruses, bacteria, and fungi, among others, sourced from the Fukushima Translational Research Project, Fukushima, Japan [21]. Briefly, during post-blocking, the microarrays were treated with diluted saliva samples along with Goat Reference Antibody Mix I (procured from Fukushima Protein Factory, Inc., Fukushima, Japan) and then stained with Alexa Fluor 647-labeled anti-human IgA and Cy3-labeled anti-goat IgG antibodies. These arrays were subsequently scanned with the GenePix 4000 B device (Molecular Devices, San Jose, CA, USA). Arrays that were untreated with any serum acted as the reference controls. To compare microarrays, the fluorescence intensity ratios (Alexa Fluor 647/Cy3) were normalized, and the relative values were calculated against the negative controls to mitigate cross-reactions with secondary antibodies. The metrics are expressed as a relative log2 ratio. A value of ≥2 was considered positive for antibodies against microbial antigens. The negative values were considered as 0.

### 2.4. Searching for Antibodies against Microbial Antigens Associated with COVID-19 Antibody Acquisition Using Machine Learning

The machine-learning model used was random forest [22], with the antibodies against the microbial antigen as the explanatory variable and the neutralizing antibody titer as the objective variable. Moreover, 80% of the data was used for training and the remainder for testing. The number of trees was 100, the loss function was “squared_error,” and the depth was not set. After training, SHAP values [23] were used to interpret the model and search for explanatory variables that would be useful in predicting the objective variable.

### 2.5. Statistical Analysis

A *p*-value of <0.05 was considered statistically significant. For comparing categorical variables, the chi-squared test was employed. We conducted statistical evaluations in Microsoft Excel 2016 (Microsoft Corporation, Redmond, WA, USA) in conjunction with Python (release 3.7.12).

## 3. Results

### 3.1. Effect of Vaccination on Antibody Profiles of Microbial Antigens

To study the changes in antibodies against microbial antigens in COVID-19 vaccinees over time through antibody profiling, eight participants were randomly selected from among those who participated in the FVCS (Table 1). Sera collected over time from these individuals were profiled for antibodies against microbial antigens using microbial protein microarrays. The number of antibodies against microbial antigens, excluding those against SARS-CoV-2, ranged from 500–900 antibodies per individual (Figure 1a, Appendix A). Although individual differences were present, no significant changes in antibody profiling after the second vaccination (three blood sample collections) or third booster vaccination (one blood sample collection) were observed (Appendix A).

The presence or absence of antibodies against individual microbial antigens was confirmed using antibody profiling data. Neutralizing antibodies against spike protein S1, which contains the RBD of the Wuhan strain and various mutant strains of SARS-CoV-2, did not show high antibody levels after the second COVID-19 vaccination (Figure 1b). In addition, we could not detect the antibodies against spike protein S1 of the Omicron strains. In contrast, antibodies against the full-length extracellular domain of the spike protein of the Omicron strains were detected at high levels. After the third vaccination (i.e., booster vaccination), the antibody levels against the Omicron strain BA.1 exceeded a fluorescence intensity ratio of 5 in all tests when compared to the negative control.

The levels of antibodies against coronaviruses other than SARS-CoV-2 were examined. Antibodies against the SARS-CoV spike protein were detected at high levels after the second vaccination, and were further elevated after the third vaccination, i.e., the booster vaccination (Figure 1c). In contrast, antibodies against the spike proteins of human coronavirus 229E, HKU1, and OC43 showed no increase in antibody levels after the booster vaccination (Figure 1d). Therefore, it is assumed that the antibodies against SARS-CoV are cross-reactive against the spike protein at the time of COVID-19 vaccination. In addition, antibodies against other viruses and bacteria were not affected by vaccination (Appendix A). In one example, antibodies to human adenovirus D were present or absent in individuals, and no effect of vaccination was confirmed (Figure 1e).

### 3.2. Relationship between Neutralizing Antibody Production and Antibody Profiles against Microbial Antigens after Vaccination

To examine the relationship between the production of neutralizing antibodies against SARS-CoV-2 and antibody profiles against microbial antigens, we selected four to eight healthy individuals who had never been infected with SARS-CoV-2 from each of the six groups classified by vaccine-elicited time-course antibody dynamics (Table 2, Appendix A). The average age of G1 (rich responders), who had high antibody titers or medium maintenance antibody titers against SARS-CoV-2, was 43 years (standard deviation of 15.5), while the average ages of G5 and G6 (poor responders), who had low antibody titers, were 59 and 70 years, respectively. Blood samples used for the analysis of antibody profiles against microbial antigens were collected between 70 and 116 days after the second vaccination.

Antibodies against microbial antigens were analyzed in 39 samples of the G1–6 groups stratified by their neutralizing antibody dynamics against SARS-CoV-2 (Table 2). The average number of types of microbial antibodies, excluding those against SARS-CoV-2, for each group, was as follows: G1 had 732 ± 180, G2 had 695 ± 210, G3 had 651 ± 116, G4 had 600 ± 107, G5 had 797 ± 225, and G6 had 742 ± 145. These results demonstrate the absence of significant differences in microbial antibody counts between the groups and the greater significance of individual differences (Figure 2a, Appendix A).

To confirm the level of neutralizing antibodies against SARS-CoV-2 in the stratified groups (G1–6), the levels of antibodies against the RBD of spike protein S1 of the Wuhan strain and various mutant strains of SARS-CoV-2 were evaluated. The results showed that G1 (rich responders), G2, and G4 had high levels of antibodies against the Wuhan strain (2–4), whereas G5 and G6 (low responders) had low antibody levels (<1) (Figure 2b). In addition, G1 and G2 had consistent levels of antibodies against the mutant strains in the delta and epsilon strains and low levels of antibodies against the other mutants. No antibodies against the RBD of the Omicron strain were detected in any of the specimens. We searched for antibodies against microbial antigens at high levels in G1 (rich responders). However, we were unable to detect antibodies present at significantly higher levels, specifically in G1.

Using data from 39 samples in groups G1–6 and SARS-CoV-2 neutralizing antibody titers measured by chemiluminescence immunoassay (Appendix A), a machine-learning search was conducted to identify antibodies against microbial antigens that contribute to the increase or decrease in neutralizing antibody titers. Antibodies against the SARS-CoV spike protein (Shapley additive explanations [SHAP] value = 101.1) and the staphylococcal complement inhibitor (SCIN; SHAP value = 42.8) were selected as antibodies that contribute to the increase in the SARS-CoV-2 neutralizing antibody titer (Figure 2c, Appendix A). Higher levels of both antibodies were positively correlated with higher levels of the neutralizing antibodies against SARS-CoV-2 (Figure 2d). In addition, antibodies against influenza B virus (SHAP value = 31.5) and Rubella virus (SHAP value = 28.6) were negatively correlated with neutralizing antibodies against SARS-CoV-2. We confirmed in our first set of samples that microbial antibodies with these high SHAP values remained stable over a 4-point blood draw (Appendix A).

## 4. Discussion

To the best of our knowledge, this study represents a pioneering endeavor to comprehensively profile microbial antigen-specific antibodies in individuals vaccinated against COVID-19. It also provides information on the history of microbial infections and the presence of antibodies against commensal bacteria in the Japanese population. For example, antibodies against the influenza virus, *Escherichia coli*, and *Staphylococcus aureus* were detected in all samples. While the quantity and diversity of antibodies targeting microbial antigens exhibited some degree of individual variation, substantial overall differences were not observed. Our study suggests that there are no significant differences in sanitation conditions in Japan, and many Japanese people have similar infection histories. This uniformity in infection and vaccination histories makes the Japanese population an interesting cohort for studying the effects of COVID-19 vaccination on antibody profiles against microbial antigens.

In the present study, there was little change in the levels of antibodies against microorganisms other than SARS-CoV-2 after vaccination. The assessment of the impact on the human immune response caused by repeated SARS-CoV-2 mRNA vaccine administration has become crucial. Several studies have discussed the induction of autoantibodies following SARS-CoV-2 vaccination [24,25,26]. As for the antibodies against microorganisms, including bacteria and viruses, the present study outcomes suggest that COVID-19 vaccination stimulates the production of highly specific neutralizing antibodies targeted at the spike protein of SARS-CoV-2 while not altering antibody profiles against other microbial antigens.

In the current study, the subjects received vaccines targeting the wild-type strain. However, in most individuals, we also observed an antibody response not just against the wild type but also against other variants, excluding the Omicron strain. This observation aligns with prior reports. Past research has consistently shown that after receiving a vaccine formulated against the wild-type strain, there remains a robust immune response to subsequent variants [27,28]. Conversely, the Omicron variant is characterized by its ability to evade immune responses, making the acquisition of a humoral immune response against it with the initial vaccines challenging [29,30,31]. Notably, in this study, while there was a robust antibody response to the primary variants, including the wild type, in all samples following the third vaccination, there was a conspicuous absence of an RBD antibody response against the Omicron variant. However, a consistent antibody response was seen against the Omicron’s non-specific spike glycoprotein. Given that neutralizing antibodies against the RBD are considered vital markers in infection prevention, the efficacy of additional vaccine doses in preventing Omicron infections might be comparatively lower than in other strains. In contrast, some reports suggest that the third booster vaccine against the wild type can indeed induce adequate neutralizing antibodies against the Omicron variant [32,33]. The testing method used in our study provides a qualitative assessment, and it is crucial to validate these findings using different testing approaches. Comprehensive measurements of antibody responses using similar techniques will likely be beneficial for future variants and COVID-19 vaccine development.

In this study, no notable differences were observed in the microbial antibody profiles of groups that were differentiated by patterns of antibody persistence after COVID-19 vaccination. Individuals can show varying responses to vaccination, with some having a more sustained antibody presence than others [4,11,12]. The reasons for these differences remain unclear. There were no apparent differences in microbial antibody profiles that could explain the cause of the variance in vaccine response. Previous studies have indicated that some Japanese individuals have a weaker antibody response to rubella and measles vaccines [34]. This suggests the presence of some constitutional predisposition common to vaccination. The microbial antibody profiles examined in this study included those against common vaccination targets in Japan, such as rubella, measles, and tuberculosis. However, no significant variations in antibody reactions to these pathogens were found. It is important to note that our study did not ascertain whether participants had previously been vaccinated against these diseases, highlighting a limitation in our research. To fully grasp the implications and reasons behind these findings, it is crucial to amalgamate more detailed information and conduct further studies.

Machine learning analysis of neutralizing antibody titers against SARS-CoV-2 and antibody profiles against microbial antigens showed a positive correlation between the increase in neutralizing antibody titers against SARS-CoV-2 and in antibodies against the spike protein of SARS-CoV after COVID-19 vaccination. The antibody profile data showed an increase in antibody levels against the SARS-CoV spike protein after booster vaccination, consistent with the machine-learning results. SCIN antibody levels were also positively correlated. SCIN is a protein secreted by *S. aureus* that inhibits C3 convertase, a central enzyme in the complement system, thereby suppressing phagocytosis and downstream effectors to counter the host immune defense [35]. Therefore, it is important for humans to develop antibodies against SCIN to prevent *S. aureus* infections. Antibodies against SCIN showed no change over time after COVID-19 vaccination. Thus, the COVID-19 vaccination might not stimulate the production of antibodies against SCIN, but rather subjects with previously high antibody levels had higher neutralizing antibodies against SARS-CoV-2. The reason that people with higher SCIN antibody levels have higher neutralizing antibody titers against SARS-CoV-2 is unclear. One theory suggests that the presence of humoral immunity against *S. aureus* indicates that the innate immune system may not rapidly clear microbial pathogens from the blood, leading to heightened adaptive immunity through B lymphocytes and plasma cells. Therefore, this could reflect the vulnerability of the innate immune system or conditions that allow commensal bacteria to easily invade the body [36]. Given the limited sample size in this study, future research should expand the number of specimens to further elucidate the relationship between SCIN antibodies and enhanced neutralizing antibody responses to SARS-CoV-2. The role of SCIN antibodies in humans also warrants further investigation.

It is widely understood that the human immune system closely interacts with resident bacteria, especially the gut microbiota, leading to discussions about the relationship between gut bacteria and the coronavirus vaccine [37,38]. The mutual interactions between gut microbiota and the immune system stabilize the latter, and disruptions in these mechanisms are known to cause immune abnormalities and diseases triggered by the induction of autoantibodies [39]. As such, the antibody profile of an individual against microorganisms might reflect their response to these gut bacteria. By examining the relationship between the formation of antibodies against the coronavirus vaccine and antibodies against resident microorganisms, one can possibly infer the correlation between gut health and vaccination efficacy. Changes in the gut microbiota have been reported after inactivated coronavirus vaccination [40]. In our study, no significant changes were observed in the overall antibody profile against microorganisms before and after vaccination. Analyzing the antibodies related to the gut microbiota more broadly may provide valuable insight into these internal changes. Investigating the link between the gut microbiota, the efficacy of the coronavirus vaccine, and the immune response to COVID-19 remains a key research priority [37]. Our current study indirectly offers valuable insight into this domain.

The findings from this research could have far-reaching implications for understanding the intricate landscape of the immune response post-COVID-19 vaccination. This pioneering endeavor to explore the relationship between antibodies against other microbes and vaccine efficacy stands out as a significant strength of our study. By highlighting the specificity of the immune response to the SARS-CoV-2 spike protein without significant alterations to other microbial antigen profiles, this study underscores the precision of mRNA vaccine platforms. Such a groundbreaking attempt not only reinforces trust in the safety and efficacy of current vaccines but also holds promise for the development of future vaccines targeting other pathogens using similar platforms. Finally, the potential of microbial protein microarrays as a tool offers a new dimension to epidemiological research, enabling a comprehensive view of individual infectious histories and immune status.

While this study offers valuable insight into the antibody profiles of individuals vaccinated against COVID-19, it is important to consider several limitations. First, the sample size, especially for particular groups, was relatively small, which could influence the robustness and generalizability of our findings. One of the strengths of this study is its ability to screen a comprehensive and broad range of microbial antibodies. However, due to the high costs involved, the number of tests was limited. Therefore, this study serves as a proof of concept for the efficacy of such a method, and there is a need to validate the results obtained in this study using methods like ELISA in a larger number of cases in future research. In addition, the study focus was primarily on the Japanese population, potentially limiting the broader applicability to diverse populations with different genetic backgrounds and exposure histories. Furthermore, while microbial protein microarrays were used for antibody detection, it remains uncertain if they capture all relevant antibodies or accurately represent the diversity of responses to different pathogens. Finally, the relationship between SCIN antibodies and SARS-CoV-2 neutralizing antibodies, although intriguing, remains correlative, and the causal relationships were not explored.

## 5. Conclusions

In conclusion, the antibody levels detected by microbial protein microarrays correlated with the neutralizing antibody titers measured by chemiluminescence immunoassay, indicating that microbial protein microarrays can be used for the comprehensive analysis of antibodies against microbial antigens produced through pathogenic infection and vaccination. In research on infectious diseases and vaccine development, antibody profiling against microbial antigens is useful for obtaining information on infection history, vaccine efficacy, and individual immune status. In addition, although the relationship between the microbiome and health maintenance and disease is known, few reports have been published on the association between antibodies against microbial antigens and health maintenance and disease. Therefore, the use of microbial protein microarrays is expected to reveal the association between antibodies against microbial antigens, and health maintenance and disease. If an association between antibodies against microorganisms and diseases is found, it could contribute to the development of disease markers and therapeutics.

## Figures and Tables

**Figure 1 vaccines-11-01694-f001:**
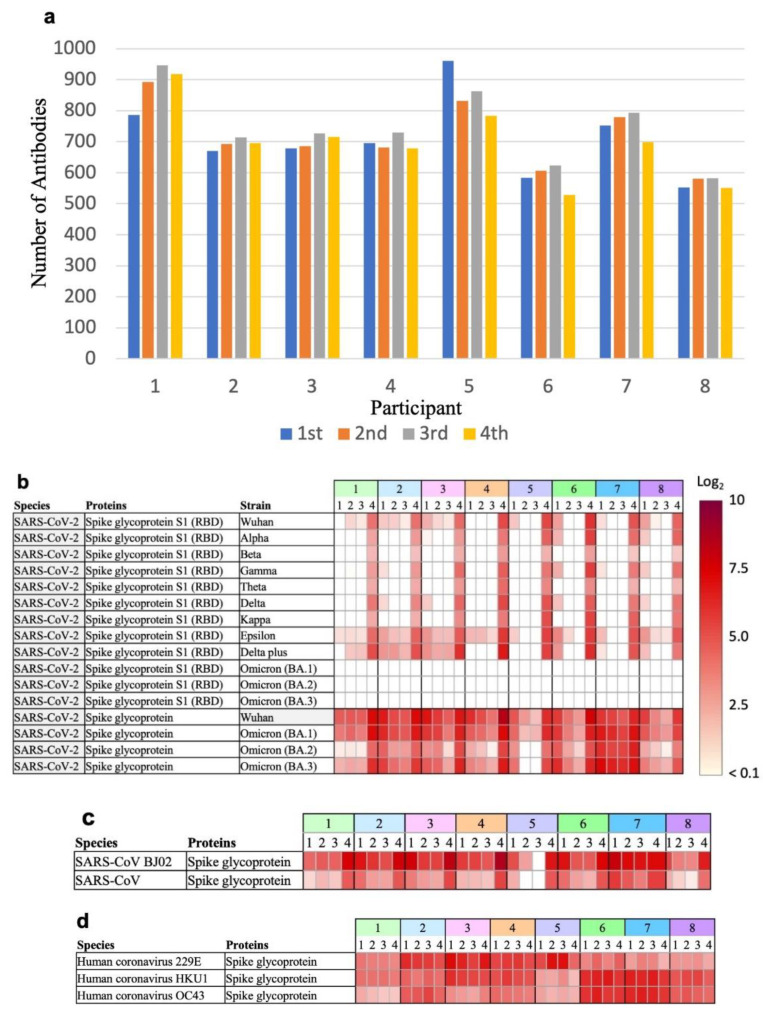
(**a**) Number of antibodies against microbial antigens. Number of antibodies against microbial antigens, excluding SARS-CoV-2 antigens, in sera collected three times (1st, 2nd, and 3rd represent the first, second, and third blood sample collections, respectively) after the second vaccination and once (4th represents the fourth blood collection) after the third vaccination from eight individuals. (**b**) Changes in the level of antibodies against spike proteins in various mutant strains of SARS-CoV-2 over time. In the Table, 1, 2, 3, and 4 represent the first, second, third, and fourth blood sample collections, respectively. The fluorescence intensity ratios of relative values against the negative controls are indicated as relative log_2_ ratios and represented by the red intensity in the figure. (**c**) Changes in the level of antibodies against SARS-CoV spike proteins over time. (**d**) Changes in the level of antibodies against human coronavirus spike proteins over time. (**e**) Changes in the level of antibodies against human adenovirus D spike proteins over time.

**Figure 2 vaccines-11-01694-f002:**
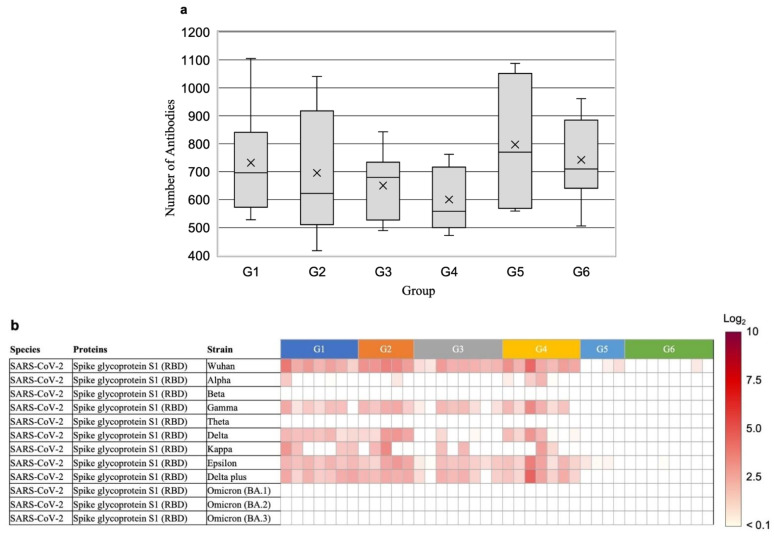
(**a**) Number of antibodies against microbial antigens, excluding antibodies against SARS-CoV-2, in the sera of six groups stratified by antibody titer and the remaining neutralizing antibodies. The cross mark represents the mean while the box represents the median, 25th percentile, and 75th percentile. The upper and lower bars represent the minimum and maximum values. (**b**) The level of antibodies against spike proteins in various mutant strains of SARS-CoV-2. The fluorescence intensity ratios of the relative values against the negative controls are indicated as relative log_2_ ratios and are represented by the red intensity in the figure. (**c**) Shapley additive explanation (SHAP) values of antibodies against each microbial antigen, indicating their level of importance. The 10 antigens are shown in order of their SHAP values. * Victoria strain; ^§^ Tokio strain; ^†^ HPV-77 strain, BR2S antigen; ^‡^ HPV-77 strain, K2S antigen. (**d**) Relationship between the SHAP value and the feature value of antibodies against each microbial antigen. The feature value of a dot is positively correlated with high (red) and negatively correlated with low (blue) values. * Victoria strain; ^§^ Tokio strain; ^†^ HPV-77 strain, BR2S antigen; ^‡^ HPV-77 strain, K2S antigen.

**Table 1 vaccines-11-01694-t001:** Characteristics of participants observed over the time course of antibody profiling.

ID ^a^	Sex	Age (Years)	Date of Blood Sampling (d)				CoV-2IgG(N) ^d^
			1st ^b^	2nd ^b^	3rd ^b^	4th ^c^	
1	Male	52	21	127	207	58	ND
2	Female	32	21	127	207	79	ND
3	Male	63	21	131	213	79	ND
4	Male	29	21	127	207	79	ND
5	Male	52	21	128	211	76	ND
6	Female	56	22	127	208	72	ND
7	Male	60	21	130	210	76	ND
8	Female	33	21	131	208	76	ND

^a^ identification number for this study; ^b^ Days from the second vaccination (d); ^c^ Days from the third vaccination (d); ^d^ IgG antibody titers against SARS-CoV-2 nucleoprotein; ND: not detected.

**Table 2 vaccines-11-01694-t002:** Characteristics of participants selected to study the relationship between neutralizing antibody titers and antibody profiles.

	G1	G2	G3	G4	G5	G6
Number of subjects (male/female)	7 (5/2)	5 (3/2)	8 (3/5)	7 (1/6)	4 (3/1)	8 (4/4)
Age (years), mean ± sd	42.8 ± 15.5	53.9 ± 13.0	48.1 ± 12.7	50.0 ± 13.9	58.5 ± 22.2	70.3 ± 13.6
Days since second vaccination (d), mean ± sd	89.1 ± 16.0	99.8 ± 7.2	80.4 ± 3.0	77.9 ± 5.1	88.8 ± 5.1	92.9 ± 5.4
CoV-2IgG(N) *	ND ^†^	ND	ND	ND	ND	ND
CoV-2Nab ^‡^, (AU/mL) mean ± sd	1299.2 ± 1286.7	659.8 ± 280.4	284.7 ± 191.5	666.2 ± 261.8	37.5 ± 2.6	15.8 ± 8.2

* IgG antibody titers against novel coronavirus nucleoprotein, ^†^ ND: Not detected, ^‡^ neutralizing antibodies against SARS-CoV-2.

## Data Availability

Data are available from the corresponding author upon reasonable request.

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
