# Peer review of "Antibody Profiling of Microbial Antigens in the Blood of COVID-19 mRNA Vaccine Recipients Using Microbial Protein Microarrays"

_vaccines, 2023, doi:10.3390/vaccines11111694_

Round 1
Reviewer 1 Report
Comments and Suggestions for Authors
Well crafted manuscript on novel subject area of overall changes ot diverse microbial antigens over time with response to SARS-CoV-2 vaccine
Changes were limited after vaccine
Interesting found staph complement inhibitor which increases neutralization
Screened 4000 microbe antigens with 3437 diverse extracts and 1606 rec proteins with serum form 8 individuals with 4 blood draws over time.
Full length spike had greater reactivity that RBD om fig 1b
2 of 8 had reactivity to adenovirus At all time points
Minor fig 1 appears twice with shapley 2nd time.’
Need to indicate tha tat 1 or 2 visit that the 9 microbes in fig 1 shaply increased over time with reactivity like RBD of SARS-Cov-2
The link for supplementary material was not working
It is customary to include supplement material with manuscript.
This was not reviewed and should be reviewed
Comments on the Quality of English Languagegood
Author Response
Reviewer 1
Comments and Suggestions for Authors
Well crafted manuscript on novel subject area of overall changes ot diverse microbial antigens over time with response to SARS-CoV-2 vaccine
Changes were limited after vaccine
Interesting found staph complement inhibitor which increases neutralization
Screened 4000 microbe antigens with 3437 diverse extracts and 1606 rec proteins with serum form 8 individuals with 4 blood draws over time.
Full length spike had greater reactivity that RBD om fig 1b
2 of 8 had reactivity to adenovirus At all time points
Minor fig 1 appears twice with shapley 2nd time.’
Need to indicate tha tat 1 or 2 visit that the 9 microbes in fig 1 shaply increased over time with reactivity like RBD of SARS-Cov-2
Reply
Thank you for your comments. We have corrected the number of the figure of sharpley.
As you mentioned, we aimed to clarify the changes in microbial antibodies after SARS-COV-2 vaccine and to evaluate the relationships between the antibody values of SARS-CoV2 and other microbial antibodies.
As for the 9 microbes with highest SHAP, Antibodies to these nine microorganisms basically showed no tendency to increase after the second or third vaccination; analysis of these antibodies in the 1st set of time-series samples showed that although there were differences in antibody responses from individual to individual, the values remains stable over the time series. This indicates that the relationship is not that the vaccine has an elevated relationship with other microbial antibodies, but that the underlying background of other microbial antibodies may associated with antibody titers against SARS-Cov2. We have added the supplementary table 5 which include the changes of the antibodies with highest SHAP over time.
Comment
The link for supplementary material was not working
It is customary to include supplement material with manuscript.
This was not reviewed and should be reviewed
Reply
Thank you for pointing out the issue with the supplementary material link. We understand the importance of including supplementary material with the manuscript. However, the provision and review of the supplementary materials are typically the responsibility of the editorial system. While it may be challenging for us to address this directly, we are willing to assist and cooperate as needed.
Reviewer 2 Report
Comments and Suggestions for Authors
The manuscript entitled “Antibody profiling of microbial antigens in the blood of COVID-19 mRNA vaccine recipients using microbial protein microarrays” by Saito et. aims to evaluate the effect of SARS-CoV2 vaccination on influencing antibodies against microbial antigens. The authors use microarray to test the samples from vaccinees for Abs against bacterial and virus antigens. The study concludes that specific SARS-CoV2 Abs were elicited via COVID-19 vaccination and the antibody profiles against other microbial antigens were not altered. The purpose of the study is unclear to me. The rationale for the authors claim that Abs against other microorganisms may be influenced by COVID-19 vaccination is not clear. Was the aim to investigate why some individual elicit neutralizing and high titer Abs vs those who don’t? It’s a far-fetched idea to relate just the microorganism Ab profiles with the qualitative and quantitative differences in COVID-19 vaccinations among individuals. This is more relevant to individual’s genetic rather than whether or not there were Abs against other pathogens.
The AZD1222 (Astrazeneca) is a replication-deficient chimpanzee adenoviral vector vaccine that expresses the SARS-CoV-2 structural surface glycoprotein antigen. It is not an mRNA-based vaccine.
Line 77-78: The authors state that “These results may be used in future vaccination strategies and vaccine development.” It’s not clear what the authors are alluding to. How do the author propose these results will benefit vaccine development. This should be stated to clarify.
Line 123: What was the rationale for collecting specimens after 90- days (~3 months) after 2nd immunization? The Abs to SARS-CoV2 are known to decline rapidly.
Line 191-192: The Abs elicited by BNT162b2 vaccine cross-reacted with Omicron strains, it would be nice to test if they can exhibit any level of neutralization. There are classes of neutralizing Abs that are not targeted to RBD but to other important epitopes on the spike.
Line 208: In the example with Abs to human adenovirus, did the individual receive AstraZeneca vaccine at any point?
The author’s report their findings as “average number of antibodies”, do they mean titer? How do they calculate the average number of Abs?
The discussion has several overstatements.
Author Response
Reviewer 2
Comment
The manuscript entitled “Antibody profiling of microbial antigens in the blood of COVID-19 mRNA vaccine recipients using microbial protein microarrays” by Saito et. aims to evaluate the effect of SARS-CoV2 vaccination on influencing antibodies against microbial antigens. The authors use microarray to test the samples from vaccinees for Abs against bacterial and virus antigens. The study concludes that specific SARS-CoV2 Abs were elicited via COVID-19 vaccination and the antibody profiles against other microbial antigens were not altered. The purpose of the study is unclear to me. The rationale for the authors claim that Abs against other microorganisms may be influenced by COVID-19 vaccination is not clear. Was the aim to investigate why some individual elicit neutralizing and high titer Abs vs those who don’t? It’s a far-fetched idea to relate just the microorganism Ab profiles with the qualitative and quantitative differences in COVID-19 vaccinations among individuals. This is more relevant to individual’s genetic rather than whether or not there were Abs against other pathogens.
Reply
Thank you for your valuable comments. Your concerns are valid. Indeed, there is significant individual variability in vaccine antibody levels, and these cannot be solely explained by the background of microbial antibodies. Our previous research also highlighted these individual differences. Past studies have suggested that these differences might be caused by variations in immune status or underlying health conditions. As you pointed out, genetic factors are also likely to play a role. However, there are many aspects of these individual differences that cannot be explained by these factors alone. In this study, we aimed to investigate whether exposure to other microbes might be related to these differences. To this end, we conducted our research on a group of healthy males and females with relatively consistent medical backgrounds and no underlying health conditions. Furthermore, we noted that the microbial antibodies we considered as the background were not influenced by the COVID-19 vaccine and did not show temporal changes. As you rightly pointed out, our sample size was small, and this study has several characteristics of a pilot study. We have added explanations regarding the relevance of related studies and the significance of our research in the introduction section, as follows:
Page 2, Lines 68–98
While COVID-19 vaccines are effective in preventing severe infections, individual differences in antibody formation exist. Moreover, there are variations in long-term antibody retention. Based on the FVCS cohort, mathematical modeling and machine learning have been used to stratify the survey participants into six groups according to the time-course patterns of their neutralizing antibody titer trajectories after their second SARS-CoV-2 vaccination [12]. In this model, G1 represented the “rich” responder group with sustained high antibody titers, while G5 and G6, the “poor” responder groups, exhibited low titers. G2, G3, and G4 were intermediate groups. Several factors have been identified as inhibitors to antibody formation post-vaccination [13]. For example, patients with immunodeficiencies, those on maintenance dialysis, and the elderly often have reduced antibody responses after SARS-CoV-2 vaccination. However, even among those without such medical conditions, variations in antibody responses exist, and the reasons remain unclear. Identifying populations with diminished vaccine efficacy is vital for determining COVID-19 vaccination priorities and discussing the need for repeated vaccinations.
The relationship between the efficacy of vaccines and infections from other microorganisms has been a topic of discussion [14]. To elucidate the individual differences in the efficacy of the COVID-19 vaccine, we hypothesize that examining antibodies against other microorganisms might offer valuable insight. This perspective gains particular relevance when considering countries like Japan, where a vast majority of the population has been vaccinated against diseases such as measles, rubella, and mumps, and where individual variations in the acquisition of antibodies from these vaccines are documented [15-17]. Furthermore, certain pathogens, such as the human immunodeficiency virus, measles virus, Salmonella, Schistosoma, and Nematospiroides, are known to exert immunosuppressive effects [18]. There is speculation on whether the presence of antibodies against these microorganisms, which suggest a current or previous infection, might serve as predictors for the efficacy of other vaccines. Furthermore, there have been instances where vaccines targeting one microorganism exhibit cross-reactivity with antigens from other closely related microorganisms [19]. In addition, the intestinal microbiota plays a pivotal role in immune responses and are thought to be associated with vaccine responses [20]. Prior exposure to microorganisms and subsequent antibody production as a result of the immune response might be associated with the efficacy of vaccines and both the humoral and cellular immune reactions [21]. Despite these speculations and preliminary findings, comprehensive studies delving into the interplay between COVID-19 vaccine efficacy and antibodies against other infectious agents remain scarce.
Comments
The AZD1222 (Astrazeneca) is a replication-deficient chimpanzee adenoviral vector vaccine that expresses the SARS-CoV-2 structural surface glycoprotein antigen. It is not an mRNA-based vaccine.
Reply
We appreciate your comment. We have removed AZD1222 from the introduction.
Comments
Line 77-78: The authors state that “These results may be used in future vaccination strategies and vaccine development.” It’s not clear what the authors are alluding to. How do the author propose these results will benefit vaccine development. This should be stated to clarify.
Reply
We appreciate your insightful comment. By elucidating individual variations in antibody levels post-COVID-19 vaccination, we can identify groups or factors associated with lower antibody responses. This understanding can be instrumental in determining vaccination priorities in the future. To clarify this, we have modified the text as follows:
Page 2, Lines 77–80
However, even among those without such medical conditions, variations in antibody responses exist, and the reasons remain unclear. Identifying populations with diminished vaccine efficacy is vital for determining COVID-19 vaccination priorities and discussing the need for repeated vaccinations.
Comments
Line 123: What was the rationale for collecting specimens after 90- days (~3 months) after 2nd immunization? The Abs to SARS-CoV2 are known to decline rapidly.
Reply
Thank you for the important observation. Previous studies have shown that vaccine antibody levels start to decline gradually after 30 days. Immediately after vaccination, there is not much difference in the decline rates between individuals. However, by 90 days, the distinction between individuals who maintain their antibody levels and those who don't becomes more pronounced. This is why we chose to investigate the relationship between neutralizing activity and microbial antibodies at this point. We added a note to clarify this rationale and also included the neutralizing activity values of these specimens in the table.
Page 3, Lines 118–129
In the second set, we examined the relationship between the production of neutralizing antibodies against SARS-CoV-2 and the production of antibodies against other microorganisms. To investigate the relationship between the production of neutralizing antibodies against SARS-CoV-2 after COVID-19 vaccination and the production of antibodies against other microorganisms, specimens that were collected approximately 90 days after the second BNT162b2 vaccination were subjected to IgG antibody profiling using microbial microarrays. We selected this timeframe because vaccine-induced antibody levels begin to vary around this period, offering a clearer insight into antibody retention in the population. We analyzed antibody profiles in six groups (G1–6, as previously described) which were stratified based on antibody formation, the temporal persistence of antibodies, and remaining neutralizing antibody levels [12]. Thirty-nine samples were selected for analysis from the six groups stratified by vaccine-elicited time-course antibody dynamics in the FVCS (Table 2) [8,12].
Comments
Line 191-192: The Abs elicited by BNT162b2 vaccine cross-reacted with Omicron strains, it would be nice to test if they can exhibit any level of neutralization. There are classes of neutralizing Abs that are not targeted to RBD but to other important epitopes on the spike.
Reply
We appreciate your valuable point. As you mentioned, it would be desirable to test whether they can exhibit any levels of neutralization. Previous studies demonstrated that, after a 2nd dose COVID-19 vaccination, people can show neutralizing antibodies even if the level of neutralization is low. In the present study, we demonstrate that the test cannot detect antibodies against spike protein S1 of the Omicron species. We have modified the relevant paragraph as follows:
Page 4, Lines 185–193
The presence or absence of antibodies against individual microbial antigens was confirmed using antibody profiling data. Neutralizing antibodies against spike protein S1, which contains the RBD of the Wuhan strain and various mutant strains of SARS-CoV-2, did not show high antibody levels after the second COVID-19 vaccination (Fig. 1b). In addition, we could not detect the antibodies against spike protein S1 of the Omicron strains. In contrast, antibodies against the full-length extracellular domain of the spike protein of the Omicron strains were detected at high levels. After the third vaccination (i.e., booster vaccination), the antibody levels against the Omicron strain BA.5 exceeded a fluorescence intensity ratio of 5 in all tests when compared to the negative control.
Comments
Line 208: In the example with Abs to human adenovirus, did the individual receive AstraZeneca vaccine at any point?
Reply
We have confirmed that these participants never received the AstraZeneca vaccine.
Comments
The author’s report their findings as “average number of antibodies”, do they mean titer? How do they calculate the average number of Abs?
Reply
We apologize for the confusion. We are referring to the average number of types of microbial antibodies that exceed the cutoff value. We have clarified this point in the text.
Page 6, Lines 234–240
Antibodies against microbial antigens were analyzed in 39 samples of the G1–6 groups stratified by their neutralizing antibody dynamics against SARS-CoV-2. The average number of types of microbial antibodies, excluding those against SARS-CoV-2, for each group was as follows: G1 had 732 ± 180, G2 had 695 ± 210, G3 had 651 ± 116, G4 had 600 ± 107, G5 had 797 ± 225, and G6 had 742 ± 145. These results demonstrate the absence of significant differences in microbial antibody counts between the groups and the greater significance of individual differences (Fig. 2a, Supplementary Table 3).
Comments
The discussion has several overstatements.
Reply 
Thank you for pointing this out. Indeed, there were several overstatements in the discussion. We have toned down our speculations to align more closely with the findings of our study.
Reviewer 3 Report
Comments and Suggestions for Authors
This is very well conducted and planned study.
Author Response
Reviewer 3
This is very well conducted and planned study.
Reply
Thank you.
Reviewer 4 Report
Comments and Suggestions for Authors
The article submitted for review is of high clinical interest as it opens the door to new ways of detecting the COVID 19 virus. It is well structured and easy to read. I would not make any modifications to it, accepting it in its current form.
Author Response
Reviewer 4
The article submitted for review is of high clinical interest as it opens the door to new ways of detecting the COVID 19 virus. It is well structured and easy to read. I would not make any modifications to it, accepting it in its current form.
Reply
Thank you. We appreciate your comment.